# Application of Scikit and Keras Libraries for the Classification of Iron Ore Data Acquired by Laser-Induced Breakdown Spectroscopy (LIBS)

**DOI:** 10.3390/s20051393

**Published:** 2020-03-04

**Authors:** Yanwei Yang, Xiaojian Hao, Lili Zhang, Long Ren

**Affiliations:** 1Science and Technology on Electronic Test and Measurement Laboratory, North University of China, Taiyuan 030051, China; 35028@llhc.edu.cn (Y.Y.); longg1994@163.com (L.R.); 2Department of Physics, Luliang University, Luliang 033000, China; greatful_zhang@163.com

**Keywords:** classification, laser-induced breakdown spectroscopy, machine learning, iron ore

## Abstract

Due to the complexity of, and low accuracy in, iron ore classification, a method of Laser-Induced Breakdown Spectroscopy (LIBS) combined with machine learning is proposed. In the research, we collected LIBS spectra of 10 iron ore samples. At the beginning, principal component analysis algorithm was employed to reduce the dimensionality of spectral data, then we applied k-nearest neighbor model, neural network model, and support vector machine model to the classification. The results showed that the accuracy of three models were 82.96%, 93.33%, and 94.07% respectively. The results also demonstrated that LIBS with machine learning model exhibits an excellent classification performance. Therefore, LIBS technique combined with machine learning can achieve a rapid, precise classification of iron ores, and can provide a completely new method for iron ores’ selection in the metallurgical industry.

## 1. Introduction

Iron ore is the main raw material for blast furnace smelting. The quality and type of iron ore is closely related to the smelting process and economic indicators. The major factor that determines the quality of iron ore is chemical composition. The accurate classification of iron is of great significance to the acquisition of raw materials in the metallurgical industry. Laser-induced breakdown spectroscopy (LIBS) is a type of atomic emission spectroscopy, which generates plasma by focusing laser pulse on the surface of sample and collects the plasma through a high-sensitivity spectrometer. LIBS has shown great potential in qualitative and quantitative analysis. It has many advantages: in-situ, no sample preparation, high sensitivity, and almost nondestructive to sample. Therefore, LIBS has developed rapidly in the past two decades. The elements can be detected whether sample is solid [1,2,3], liquid [4,5] or gas [6]. LIBS has been applied to various fields: environment [7,8,9], biomedical [10,11,12], industry [13,14,15,16], and agriculture [17,18,19]. In addition, LIBS is also used for examination of band emissions from small molecules and for detection of organic molecules [20,21,22,23,24,25]. Mining the acquired spectral data is a hot trend for future LIBS application.

At present, more and more machine learning algorithms are combined with LIBS to perform more accurate classification, regression, clustering, and other operations on samples. Elli Bellou et al. propose an efficient, ultrafast method for olive oil classification that combines LIBS and machine learning algorithms for retrieving the spectrum information, and the accuracy rate is up to 99.2% when Linear Discriminant Analysis (LDA) algorithms are applied to build the prediction model [26]. Haobin Peng et al. provide a novel technology that can deal with the classification of substances such as coal, municipal sludge, and biomass, and results show that the accuracy of hybrid classification model is over 98% when K-means and support vector machine (SVM) work together [27]. Daniel Diaz et al. apply LIBS and principal component analysis (PCA) to the classification of gold ores prepared as pressed pellets from pulverized bulk samples. The results demonstrate that the ability of PCA to identify the Au emission line is highly related to LIBS spectral width. The performance of PCA is best when taking Au emission line as center and 0.15 nm as spectral range. At the same time, the size of gold particles also has a great influence on the acquisition of the raw spectral data [28]. Xiaohui Li et al. put forward a multivariate statistical method combined with LIBS to complete the discrimination of soft tissues like fat, skin, muscle, and so on. The k nearest neighbor (KNN) and SVM are used to build the model, and 10-fold cross validation is used to evaluate its robustness and accuracy. The results show that the accuracy is over 99.83% and sensitivity is over 0.995 when using KNN and SVM classifiers. The discrimination performances of highly similar ham, loin and tenderloin muscle are acceptable when SVM kernel function is adopted [29]. Sha Wen et al. proposed the method of combining LIBS and support vector regression (SVR) to detect three elements of nitrogen, phosphorus, and potassium in fertilizer. The results show that LIBS coupled with the least squares-support vector regression (LS-SVR) can accurately and quantitatively analyze elements in complex matrices [30]. Liwen Sheng et al. applied the combination of LIBS and random forest (RF) for the identification and discrimination of iron ore. The results show that the average prediction accuracy of the RF method can reach 100% [31], but the construction of the RF method is usually very complicated and easy to cause overfitting. Ping Wang et al. propose a method of LIBS technique coupled with variable importance measures-random forests (VIM-RF) to accomplish the analysis of acidity of iron ore. The obtained results show that the VIM-RF model has better performance than partial least squares (PLS) and least squares support vector machine (LS-SVM), with root mean squared error (RMSE) = 0.0554 wt% and *R*^2^ = 0.9103. A similar approach is also used to detect the toxic elements in polypropylene and can achieve real-time monitoring of plastic trash [32,33]. Yun Zhao et al. developed a method based on LIBS and deep belief network (DBN) to detect soil contaminated with Pb. At different levels of Pb, the results show that deep learning can handle LIBS data and highlight the importance of using samples. The total accuracy values are 98.47% (training set) and 90.625% (test set) for samples [34].

In this study, the combination of LIBS technology and different machine learning algorithms is used to complete rapid and precise classification of various kinds of iron ore. At first, the spectral data of 10 iron ore were obtained by LIBS, using PCA algorithm to reduce the dimensionality of raw spectral data. Then KNN, SVM and neural network algorithms were used to classify the ore. Finally, the accuracy and robustness of three models were evaluated.

## 2. Experiments and Methods

### 2.1. Instrumental Setup

A schematic diagram of traditional LIBS experimental setup is shown in Figure 1. In air, the laser used here is Nd: YAG laser (type: Quantel Ultra 50; wavelength: 1064 nm; repetition rate: 10 Hz; pulse duration: 10 ns; pulse energy: 200 mJ) from Quantel. The emission of the plasma was collected by using a one-inch diameter quartz lens and is introduced to an optical fiber which is couple to a seven-channel wide-spectrum spectrometer (AvaSpec-ULS2048-7-USB2 from Avantes, Netherlands). At the same time, the spectrometer is equipped with a 2048 pixels CCD detector, covering the spectral range from 190 to 950 nm with a resolution of 0.05 nm, which can be triggered by an external TTL level, or an output level trigger external device. The focusing lens is 36 mm in focal length to focus the pulse on sample surface. The sample is placed on a sample stage with freely controllable *XYZ* axis. The stroke of the stage is 5 cm and this can realize the micron level (<1 um) high precision positioning control.

The ten iron ore samples used in the experiments are from Wuhan Veteran Geological Science Teaching Instrument company: magnetite, cobalt-bearing magnetite, hematite, oolitic hematite, mica hematite, maghemite, pyrite, pyrrhotine, siderite, and limonite. They are all primitive stones without any chemical treatment, so they can be considered to have a distinct geographical representation. We chose the relatively flat side of the ore, fixed it with plasticine, and placed it on the sample stage for laser-induced breakdown after continuously adjusting parameters, operating at wavelength of 1064nm with an energy of 50 mJ, spot size after focusing of 75 um, repetition rate of 2 Hz, acquisition delay time of 1us, and integration time of 10 ms. Five pulses are shot on the sample to clean the oxides and other impurities on the surface to ensure similar conditions. At the same time, the entire system is stable after the instrument is started, changing the sample ablation point by controlling the position of sample stage. A 3 × 3 matrix was ablated on the surface of each ore. Each point of the matrix was ablated five times, and the data of each time is recorded. 45 independent spectral data were collected on each ore for subsequent analysis. The repeatability of the data is acceptable. All data is randomly split into two parts: training set consisting of 70% (315 spectra) of the data and the rest 30% (135 spectra) of the data is used as test set.

### 2.2. Methods

The open-source machine learning Python library Scikit-learn and deep learning framework Keras facilitate the analysis of data. The tools can implement and verify multiple algorithms such as PCA, SVM, RF. Firstly, the raw data has a dimension of 12,248. If the dimension is not reduced, all information in the spectrum can be reserved, but this approach will create data redundancy and encounter huge calculation tasks. Accordingly, it is indispensable to reduce the dimension of data properly. PCA is employed for reduction of the dimensionality of raw data, as an unsupervised method of dimensionality reduction. Considering the covariance of all data, the principal components of data are determined by calculating the eigenvalues and eigenvectors of covariance matrix [35]. Second, KNN, SVM, and neural network are constructed to accomplish the classification of samples. KNN algorithm classifies the samples by calculating the distance between the point to be classified and all samples, specifying the K value, and using label information [36]. As one of the most robust and accurate methods in data mining, SVM is classified by finding a hyper-plane which can differentiate the classes correctly and has the largest geometric spacing. It can only solve the linear two-classification problem initially. With the introduction of kernel function, multiple SVMs are stacked to get through complicated multi-classification tasks [37]. As the most basic structure in the deep learning field, DNN classifies samples by setting activation function, loss function, etc. [34]. Theoretically, DNN can solve any classification problem. Finally, the classification accuracy of the above algorithms is evaluated and visualized by calculating confusion matrices. 

## 3. Results and Discussion

### 3.1. LIBS Spectra of Iron Ore

Since the spectrum of other ore is too dense, the spectrum of pyrite is used to represent the general elements and it is shown in Figure 2a. According to the National Institute of Standards and Technology (NIST) [38], the nonmetallic elements (C, H, O, N, S) and metallic elements (Mg, Ca, Mo, Fe) were observed.

All spectral lines of 10 kinds of ores are shown in Figure 2b. We can see that there are obvious differences in the spectral intensity between different ores. For example, the intensity of the iron element excited in cobalt-bearing magnetite is much greater than that of the iron element in magnetite, but the presence of carbon can be clearly detected in magnetite. For hematite, oolitic hematite, mica hematite, and maghemite, the types of element are roughly the same. It should be noted that the intensity of mica hematite spectrum at Al 308.702 nm and Na 589.592 nm is significantly higher than the other three. For the remaining four kinds of ores, their spectral lines are not as dense as before, especially pyrite. The excited spectral lines are very sparse. At Mg 278.141 nm, the intensity of the line of pyrite is much greater than the intensity of the line of pyrrhotite. At K 766.489 nm, the intensity of both pyrite and limonite spectrum is basically undetectable, indicating that there is no potassium. All in all, the existence of some of these features makes the classification algorithm operational.

### 3.2. Dimension Reduction with PCA

The data we obtained is a total of 12,248 spectral points from 190 nm to 950 nm. Each data point contains information about the type of ore, but a too massive data dimension will cause difficulties in calculation. It is necessary to reduce the dimension of data appropriately. PCA is a technique for analyzing and simplifying data sets and is often used to reduce the dimensionality of a data set, while maintaining the features that have the largest variance contribution. This is done by preserving low-order principal components and ignoring high-order principal components. Such low-order components can retain the most important aspects of the data. After standardizing the obtained spectral data, PCA is carried out to reduce the dimension and transform the whole spectrum into multiple principal components. Through the PCA of the ore spectrum, the interpretation rate and cumulative interpretation rate of the top 10 principal components are obtained. The interpretation rate of the 10 principal components are 44.46%, 25.29%, 9.02%, 5.19%, 2.36%, 1.89%, 1.73%, 1.51%, 1.06%, and 0.93%. The result is shown in Figure 3. 

It can be seen from Figure 3 that the sum of the contributions of the top 10 principal components of ores reached 93.42%, indicating that these are enough to cover most of the ore spectrum information. Figure 4a and Figure 4b respectively showed the score chart of the first two principal components and the first three principal components.

Each point represents a set of data. It can be seen from Figure 4a,b that characteristic points of the same type of ore appear as obvious aggregation. Cobalt-bearing magnetite was found to be clearly separated from the other ores, which could be attributed to the fact that its spectral intensity and species are significantly higher than other classes. For the remaining nine ores, the data points distributed in space were mixed to a greater or lesser extent. Magnetite, pyrite, siderite, and limonite could be distinguished in 2D space. The difference between pyrite and pyrrhotite was also obvious. On the contrary, the distribution of mica hematite and hematite in 2D space basically coincided, indicating that the characteristics of the two ores were similar. Both maghemite and oolitic hematite were widely distributed, indicating that the composition of these two ores was more complicated.

### 3.3. Classification with Machine Learning Techniques

It can be shown from the above that, after PCA dimension reduction of the data, the top 10 principal components can roughly represent all information of the spectra. These components were used as input features for classification. First, KNN algorithm was implemented on training dataset. The distances from the 5 nearest points to the test points were calculated and stored in an array, picking the majority class of array and using it as the category of test point. The distance metric used in KNN model is Euclidean distance. The corresponding confusion matrix for KNN model is presented in Figure 5. The accuracy of KNN model is 82.96%.

Secondly, a neural network model was built to improve the accuracy of classification. The number of layers is 3, and the number of nodes in each layer is 12, 8, 10 respectively. The activation function used by the first two layers is the “Relu” function, and “SoftMax” is used as the activation function of the last layer. The DNN model’s loss function is “cross entropy” and the objective function is optimized using the “Adam” optimizer. The accuracy of training set and test set were 100% and 93.33%. The results showed that the neural network model can clearly distinguish all data in training set, and it can achieve good results in test set. The confusion matrix of neural network model is shown in Figure 6.

Finally, a SVM model with linear kernel was used for classification. The penalty factor is 1.5. The discrimination accuracy of the model on training set was 98.7%, but the discrimination accuracy on test set was 94.07%. For this reason, the discrimination performances were better than in the above two models. The corresponding confusion matrix for SVM model is presented in Figure 7.

The results showed that the combination of LIBS and machine learning algorithms can improve the classification of iron ore. Iron ore with similar appearance, but with very different actual composition, can be almost completely discriminated. The classified iron ore can be used in different fields of industry to improve its utilization rate. For example, hematite is used to make pigments, pyrite is an important raw material for making sulfuric acid, and magnetite is used for steel smelting. Therefore, precise classification could maximize utilization efficiency. During the experiment, we did not use complex pre-treatment on iron ore, such as crushing, grinding, etc. The roughness of the ore surface will affect the experimental results to a certain extent, but the data processing without pre-treatment also retains as much of the original characteristics of iron ore as possible, and showed that this technology can be used for fast real-time detection.

## 4. Conclusions

The combination of LIBS and machine learning algorithm has been used for the classification of iron ore. LIBS spectra of 10 original iron ores were acquired by LIBS technique, and the accuracy of KNN model, neural network model and SVM model are 82.96%, 93.33%, and 94.07% respectively. The obtained results sufficiently demonstrated that LIBS coupled with machine learning algorithm is a practical technique for rapid and on-line analysis of categories of iron ore and can provide a new method and technology for selection and quality control of iron ore in the metallurgical industry.

## Figures and Tables

**Figure 1 sensors-20-01393-f001:**
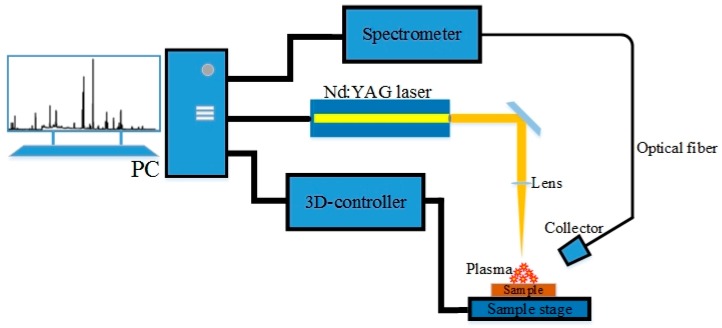
Schematic diagram of the experimental setup.

**Figure 2 sensors-20-01393-f002:**
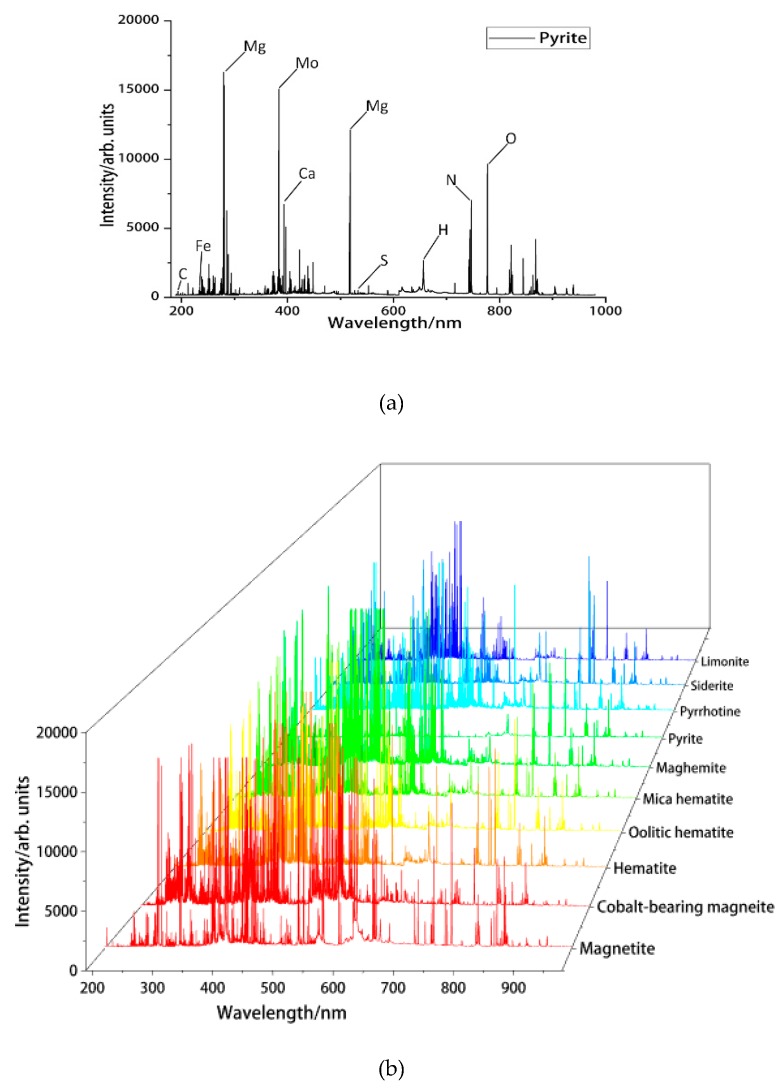
(**a**) Characteristic plot of pyrite; (**b**) Characteristic spectrum of ten iron ore.

**Figure 3 sensors-20-01393-f003:**
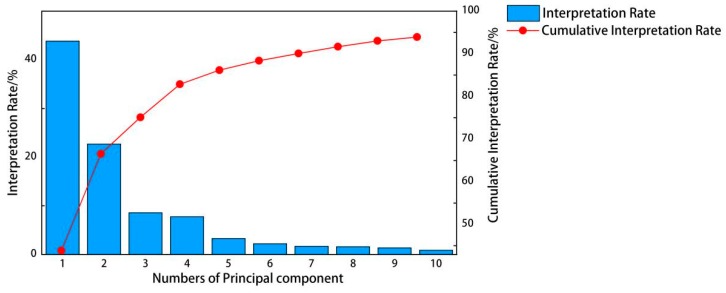
Each principal component interpretation rate and principal component cumulative interpretation rate.

**Figure 4 sensors-20-01393-f004:**
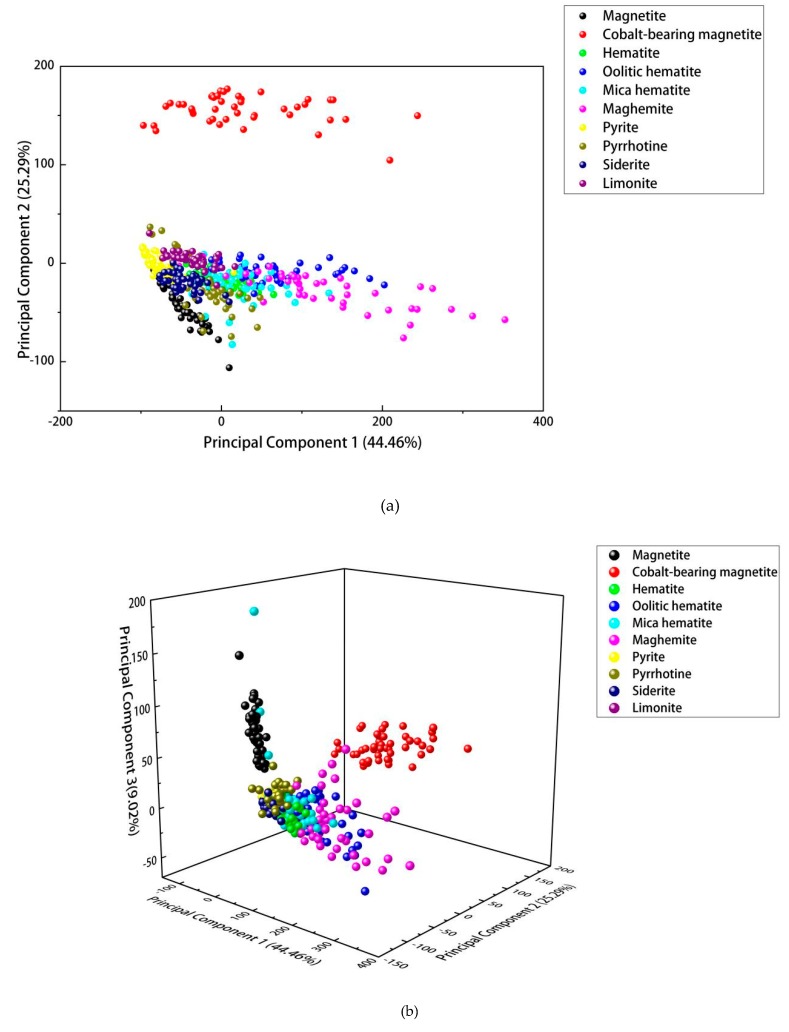
(**a**) 2D plot for the first two principal components; (**b**) 3D plot for the first three principal components.

**Figure 5 sensors-20-01393-f005:**
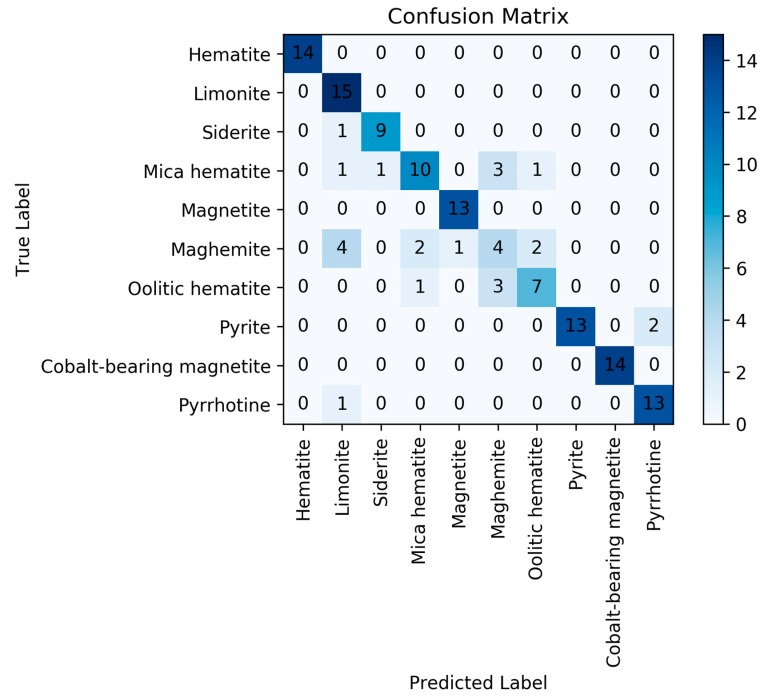
Confusion matrix for the k nearest neighbor (KNN) model.

**Figure 6 sensors-20-01393-f006:**
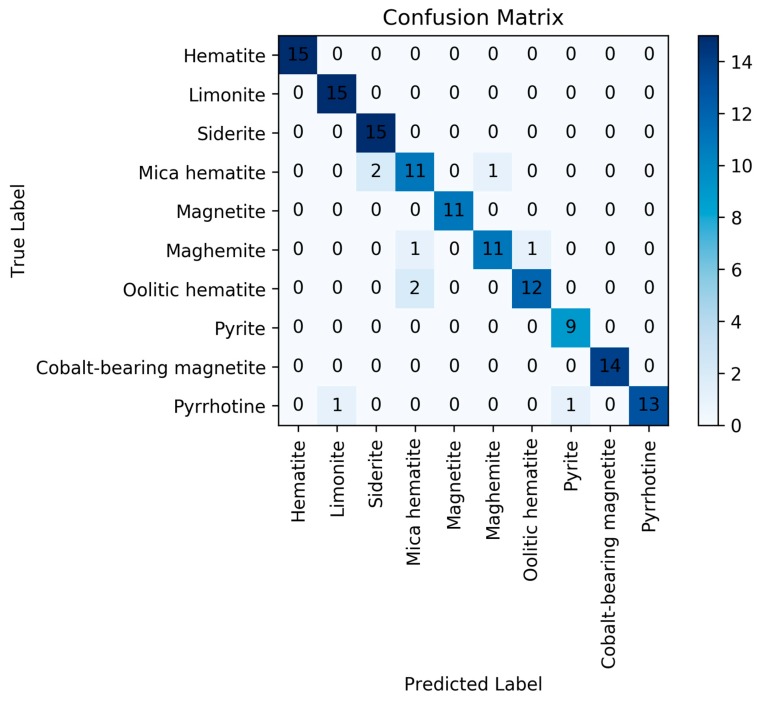
Confusion matrix for neural network model.

**Figure 7 sensors-20-01393-f007:**
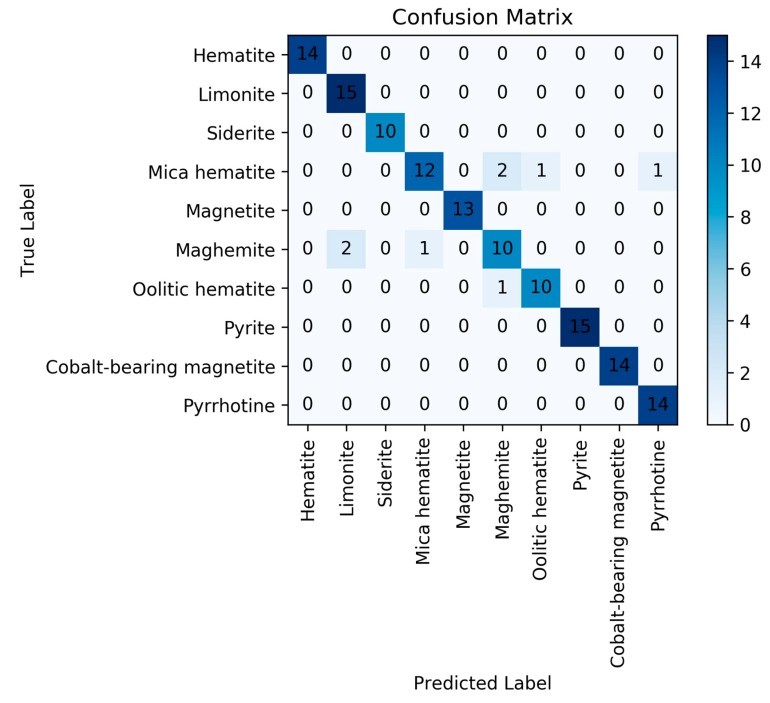
Confusion matrix for the support vector machine (SVM) model.

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
