# Peer review of "Application of Scikit and Keras Libraries for the Classification of Iron Ore Data Acquired by Laser-Induced Breakdown Spectroscopy (LIBS)"

_sensors, 2020, doi:10.3390/s20051393_

Round 1

Reviewer 1 Report

This manuscript refers to the use of laser-induced breakdown spectroscopy (LIBS) combined with machine learning methods for iron ore classification. The authors addressed most of the points raised in the previous review, but yet some issues still have to be addressed.    

-           In the first paragraph, it is mentioned that plasma emission is collected through a high-resolution spectrometer.  This is not correct – your spectrometer is not of high resolution.

-           The authors mention now that the LIBS is also used for detection of organic molecules, but referred only to studies of J. Laserna.  It would be worth to refer to previous studies on this issue by I. Bar, S. V. Rao, A. W. Miziolek and their groups, etc...

-           In the experimental section

Some more details regarding the measurement were given, however, it is still not completely clear how was the measurement performed.  What is meant by the repeatability of the data is acceptable?

-           A reference to the NIST data base should be given. 

-           I meant to show all three matrices, as different panels of a figure.

-          In the Conclusion it is mentioned the accuracies of the different were 92.96, 96.33, 98.7%.  It seems that the first number is wrong.  Why is the last number less accurate?  Check that the same numbers are given across all the manuscript.

Reviewer 2 Report

Major comments

The presented paper deals with a classification of iron ores by means of machine learning algorithms. The authors utilized three methods, namely k nearest neighbours (KNN), deep neural networks (DNN) and support vector machine (SVM) algorithms to classify ten iron ores samples. Prior to the classification, the dimensionality of the dataset was reduced by principal component analysis (PCA). Afterwards, the first ten PCs were introduced to the above-mentioned tools (KNN, DNN and SVM). The success rate of classification for KNN, DNN and SVM was 92.96%, 96.33% and 98.7% respectively.

Introduction

In my opinion, the introduction is insufficient. There are missing some important references, e.g. Sheng et. Al. 2015 DOI: 10.1039/C4JA00352G is dealing with almost the same problem (and you can find many others). Moreover, using phrases such as “As is known to all…” in a scientific paper is more than inappropriate. At least you should add references proving your statement. If you are listing LIBS applications please refer the review articles primary.

Experiments and Methods

Instrumental Setup

The description of the instrumental setup is unclear. It starts with a description of an experimental setup followed by a description of samples and experimental conditions. In addition, there are information missing: what type of Quantel laser did you use, what kind of optics was used for focusing laser beam (including focal length)? How did you collect the plasma radiation? The statement “…the plasma was collected by seven-channel wide-spectrum spectrometer…” is misleading. After the five cleaning shots how many measurements were performed on each spot?

Methods

Please change the reference [25] for PCA in LIBS, a review 2018 (https://doi.org/10.1016/j.sab.2018.05.030)

Results and Discussion

LIBS Spectra of Iron Ore

Please add a reference to the NIST (Example of how to reference these results: Kramida, A., Ralchenko, Yu., Reader, J., and NIST ASD Team (2019). NIST Atomic Spectra Database (ver. 5.7.1), [Online]. Available: https://physics.nist.gov/asd [2020, February 11]. National Institute of Standards and Technology, Gaithersburg, MD. DOI: https://doi.org/10.18434/T4W30F)

Please omit the first sentence on the page 4: “From the intensity of elements, the approximate elemental composition of the ore was expressed”. There is no intensity of elements but spectral lines.

Figure 2b is worthless. It is impossible to recognize any real differences in the spectra in this kind of visualization. For pointing out the differences of spectra of selected samples much more valuable will be the loadings of the first few PCs.

The presented manuscript provides a direct comparison of 3 machine learning tools for classification of samples with relatively complex LIBS spectra, which is potentially interesting for the sensors readers. However, the scientific novelty is very low and I will recommend to accept the paper, after addressing all the comments above, as a technical note rather than full paper.

Reviewer 3 Report

The manuscript is interesting. It deserves publication, after major revision.

There are some problems with the English usage. I think that the meaning is always clear, but many sentences look very odd.

The contribution of the manuscript would be more important if the iron concentration in ore could be determined by using of machine learning, not just classification of ores.

It seems that 45 spectra used for each ore are randomly chosen, so this randomness is a part of determined principal components. Is it good, because it helps classification, or is it bad, because random information is built-in? The authors should address this point.

A few more sentences about used Avantes spectrometer will be welcomed.

There is no sufficient information for the reader to repeat the experiment. Are some groups of the calculated ten principal components (their scores?) used as training data set for KNN? And other scores were used to perform the classification? What about data preprocessing?

I use SOLO software package (Eigenvector research); I admit that I am not familiar with the software used by authors, but more information about used methods will be welcomed.

Reference [22] is not about KNN – it is about using PCA for classification of LIBS data. I believe that [23] is also wrongly cited…. And so on…. Authors should check the reference numbers.

Round 2

Reviewer 1 Report

This manuscript refers to the use of laser-induced breakdown spectroscopy (LIBS) combined with machine learning methods for iron ore classification. This approach was used in previous studies as well. Although the authors addressed some of the points raised in the previous review, some issues still have to be addressed.  In addition, the English has to be improved.    

-           In the Abstract and through the text, the accuracy of the models with different number of significant digits is given.  Is the accuracy of the support vector machine model less than that of the other models?

-           The authors added Refs. regarding the molecular emission, but Refs. 23 and 24 are not so relevant.  The authors should rather refer, for instance, to:

  1. L. Gottfried, F. C. De Lucia, Jr., C. A. Munson, and A. W. Miziolek, Strategies for residue explosives detection using laser-induced breakdown spectroscopy, J. Anal. At. Spectrom. 23, 205–216 (2008).
  2. Portnov, S. Rosenwaks, and I. Bar, Emission following laser-induced breakdown spectroscopy of organic compounds in ambient air, Appl. Opt. 42, 2835-2842 (2003).
  3. Portnov, S. Rosenwaks, and I. Bar, Identification of organic compounds in ambient air via characteristic emission following laser ablation, J. Lumin. 102, 408-413 (2003).

-           In the introduction, R2 = 0.901 is mentioned – should it be R2?

-           In the Experimental, there is no need to give the specifications of the laser, but rather its model and operation parameters.

-           Following the definition of abbreviation, the abbreviation should be used and not the full name – for example, on p. 3, laser-induced breakdown.  In the methods – random forest should not be defined again.

-           On p. 5 line 5, it is written After standardizing…  what is meant by that?

-          On p. 5, principal component, PCA analysis are used – PCA already includes analysis.    Again, the abbreviation should be used.  

Reviewer 3 Report

The manuscript is interesting. It deserves publication, still after major revision. The authors didn't take seriously major revision of the text, they just provided the answers to comments, keeping the improvement of the manuscript as small as possible.

I still think, repeat: "There is no sufficient information for the reader to repeat the experiment. Are some groups of the calculated ten principal components (their scores?) used as training data set for KNN? And other scores were used to perform the classification?" To make it more clear, what is size of training set, what is size of test set? Where are the loadings?

Many readers would be interested to improve their LIBS (me too) by machine learning. However, authors provide many general information about ML, but very little information about what they have done, just obtained results were presented.

The confusion matrices look somehow unusual. If 10 ores were analyzed, each with a 45 spectra, then the reader would expect that all ores will be equally represented in test set, for example with 15 spectra, assuming that 30 spectra were used for training. However, the number of spectra are different between ores, moreover, they are different between various ML techniques. Why? Are some "non-representing" spectra removed from test to obtain better results?

The accuracy of matrix in Fig 7 is 0.94, not 98.7%. I have uneasy feeling that some kind of "post-processing" of results was done by authors.

What ore is shown in Fig 3? Or, is it a PCA of all training data? In text, it says "contribution rate", on plot it is "interpretation rate", figure caption says scores....

Round 3

Reviewer 3 Report

The author’s reply confirmed my suspicion that authors have only superficial understanding of PCA – because of that they slightly confuse terms (scores, don’t know what are loadings, etc), confuse accuracy of training set and test set, randomness.

However, they efficiently use Scikit and Keras. So, my suggestion is to publish this manuscript, now after minor revision. The title should be something like:

“Application of Scikit and Keras libraries for the classification of iron ore data acquired by Laser-Induced Breakdown Spectroscopy (LIBS)”

And, maybe a few sentences more in Introduction and Methods sections if looked appropriate.

Now, it should be pointed out that there is a general problem with using software: if input data are in appropriate format, the code will always provide some output data. In some cases, the output data will have no meaning. I believe that this is not the case here.

Author Response

This manuscript is a resubmission of an earlier submission. The following is a list of the peer review reports and author responses from that submission.

Round 1

Reviewer 1 Report

This manuscript refers to the use of laser-induced breakdown spectroscopy (LIBS) combined with machine learning methods for iron ore classification. Following reduction of the dimensionality of the spectral data by principal component analysis algorithm, different models were applied for spectroscopic data treatment and classification. Different accuracies were obtained by the employed models and classification could be performed.  The authors came to the conclusion that this approach will lead to iron ores classification and will assist in selecting iron ores for the metallurgical industry.  The authors should address the following issues, before the manuscript can be considered again.    

-           In the first paragraph, it is mentioned that LIBS is an atomic emission spectroscopy and is used for elements detection.  It should be worth mentioning that LIBS is related to photons emitted by plasmas, resulting from laser ablation of materials and references should be given.  Furthermore, LIBS is not used just for elemental analysis, but also for examination of band emissions from small molecules and for detection of organic molecules.   It would be worth to refer to works in the field, of J. Laserna, I. Bar, S. V. Rao, A. W. Miziolek and their groups, etc...

-           In the experimental section

- the authors should mention whether the system was stabilized before data collection. - what integration times for the spectrometer were used?

- was the sample moved between measurements?

- It is mentioned that 45 independent spectral data were collected on each ore – the authors should mention, or may be show what was the repeatability.

-           In Fig. 2 – it can be hardly seen which elements and ores are written. The font size should be increased.   In addition, a.u. should be changed to arb. units.

-           Below Fig. 2, wavelengths of Al 308.702 nm and Na 589.592 nm and other elements are given.  What is the spectrometer accuracy?   References for the identification of the lines should be given.

-          In the first line of Sect.3.2 the wavelength range of 188 to 990 nm is given, but it is different than that given in the experimental, 190-950 nm.

-           References to PCA and other methods should be given.

-          The sentence “After classifying different types of iron ores, the iron ore can be used more efficiently” should be corrected.

-          The issue of pretreatment on the iron ore was not tested, but the authors think that the original characteristics of the ore would be retained – this remains to be examined and is not proved here.

Might be worth to present the three confusion matrices in one fig.

Reviewer 2 Report

The paper submitted by Yang et al. starts from an incorrect statement, i.e. that iron ore classification by LIBS is problematic and suffers from low accuracy. In fact, the analysis of geological materials is one of the emerging fields in which LIBS (and machine learning techniques) are obtaining excellent results. The manuscript, in fact, does not show any novel result or approach. Moreover, the experimental design and data analysis is poor and basically flawed: only 10 samples are considered for the analysis and, as far it can be understood from the manuscript, the test samples were selected among the samples used also for validation and training, which is an error. At the end, one should stress that the final results, which determines SVM approach as more effective than Artificial Neural Network, is based on the fact SVM approach to misclassify 2 spectra while ANN fails for 5 spectra (over about 150 test spectra!). The difference between the performances of the two methods is, thus, statistically irrelevant.

Reviewer 3 Report

The authors used three machine learning techniques namely K-nearest neighbor, artificial neural network and support vector machines to classify ten different iron ore samples after dimension reduction with principal components analysis. They achieve very good classification accuracies.

Overall, the work was good but before it can be published, there needs to be revisions by the authors in certain sections of the manuscript to bring about clarity and understanding.

There are typographical and grammatical l errors in the manuscript that makes reading and understanding difficult in certain sections.

 A case in point is the authors describing one of the advantages of LIBS as “in-suit”. I am not sure if they meant to say ‘in-situ”.

Also, in the first paragraph of section 2.1, the authors say “…the stroke of stage is 5cm and can realize the micron level…”. I do not understand what they mean by this statement.

The last sentence of the second paragraph in section 3.1 states that “All in all, the existence of so many features makes the classification algorithm operational.” This statement seems to suggest the presence of many features is beneficial for the classification, but they contradict this very statement in the next section (3.2) by doing a dimension reduction. I suspect the statement in 3.1 is grammatically incorrect and should be corrected.

Lastly, in the last paragraph of section 3.3, the statement “This technique could be used for Industrial smelting, different iron ore had different functions.” is unclear. I have no idea what the authors are trying to say.

The description of the classification techniques (i.e. KNN, DNN and SVM) are inadequate. The authors need to give the parameters used for each technique so other researchers can independently reproduce their work. For example, the number of nearest neighbors and the distance metric used in the KNN should be specified.  For DNN, the number of layers, number or nodes per layer, activation functions, loss function, and optimization method should all be described classification results are known to depend heavily on the hyperparameters.  Similarly, parameters should be given for SVM.

Only 18 out of the 25 references are indicated in the text. This should be fixed.  The geological institute from which the iron ores were obtained should stated explicitly.

In section 2.2 the authors state “the classification accuracy of above algorithms are evaluated by using 10-fold cross validation, the LIBS data were split in two part randomly: the train set accounts for 70% of the total data and the rest 30% is used to test the models’ accuracy.” This statement is confusing because it is not clear how they combine the 10-fold cross validation and the 70-30 split.

In section 3.1 the authors state that “From the intensity of elements, the approximate elemental composition of the ore was expressed.” However, I do not see anywhere in the manuscript of where they indicate elemental compositions. This should be fixed.

Round 2

Reviewer 2 Report

In their reply to my original comments, the authors confirmed my opinion that the manuscript is not suitable for publication: "Due to the lack of sufficient statistical theory, the results cannot be fully explained, and further research will be conducted later." The paper should be eventually resubmitted after the additional work that the authors are planning to condict, with the aim of improving the level of confidence of their analysis.